# The EEG-Based Fusion Entropy-Featured Identification of Isometric Contraction Forces under the Same Action

**DOI:** 10.3390/s24072323

**Published:** 2024-04-05

**Authors:** Bo Yao, Chengzhen Wu, Xing Zhang, Junjie Yao, Jianchao Xue, Yu Zhao, Ting Li, Jiangbo Pu

**Affiliations:** 1Institute of Biomedical Engineering, Chinese Academy of Medical Sciences and Peking Union Medical College, Tianjin 300192, China; aragon_y@outlook.com (B.Y.); wuchengzhen214@foxmail.com (C.W.); xin_zhang_bme@163.com (X.Z.); yaojie011026@gmail.com (J.Y.);; 2School of Life Sciences, Tiangong University, Tianjin 300387, China; 3Department of Orthopedic Surgery, Peking Union Medical College Hospital, Chinese Academy of Medical Sciences and Peking Union Medical College, Beijing 100730, China; xuejianchaofj@163.com (J.X.);

**Keywords:** EEG, entropy, isometric contraction, fusion entropy

## Abstract

This study explores the important role of assessing force levels in accurately controlling upper limb movements in human–computer interfaces. It uses a new method that combines entropy to improve the recognition of force levels. This research aims to differentiate between different levels of isometric contraction forces using electroencephalogram (EEG) signal analysis. It integrates eight different entropy measures: power spectrum entropy (PSE), singular spectrum entropy (SSE), logarithmic energy entropy (LEE), approximation entropy (AE), sample entropy (SE), fuzzy entropy (FE), alignment entropy (PE), and envelope entropy (EE). The findings emphasize two important advances: first, including a wide range of entropy features significantly improves classification efficiency; second, the fusion entropy method shows exceptional accuracy in classifying isometric contraction forces. It achieves an accuracy rate of 91.73% in distinguishing between 15% and 60% maximum voluntary contraction (MVC) forces, along with 69.59% accuracy in identifying variations across 15%, 30%, 45%, and 60% MVC. These results illuminate the efficacy of employing fusion entropy in EEG signal analysis for isometric contraction detection, heralding new opportunities for advancing motor control and facilitating fine motor movements through sophisticated human–computer interface technologies.

## 1. Introduction

Upper extremity movement, encompassing all forms of motion involving the arms, hands, and fingers, is a cornerstone of human functionality. This movement allows us to engage in a wide range of daily activities, from the ordinary to the highly skilled. The flawless execution of these movements depends on a complex coordination network between the brain and the muscular system. The coordination involved in upper extremity movement extends beyond just moving a limb from one place to another. This requires a delicate balance of maintaining the right posture, moving in the desired direction, and exerting the necessary amount of force. This delicate balance is crucial for tasks requiring a high level of skill. Effectively grasping objects of various shapes and sizes, writing legibly and with precision, and performing delicate tasks, such as those found in detailed craftsmanship or surgical procedures, all depend on this precise coordination.

The essence of understanding how the brain orchestrates these precise movements lies at the heart of advancing our approaches to rehabilitation. For individuals who have suffered motor impairments due to events such as strokes or traumatic brain injuries, this knowledge is particularly transformative. It opens the door to developing more effective rehabilitation techniques that are tailored to the unique ways in which the brain controls movement. By focusing on these mechanisms, therapists and medical professionals can improve treatment protocols and enhance the quality of life for those affected by such impairments. 

In recent years, the field of neurotechnology has made significant progress in understanding how to connect brain function with motor recovery. One of the most promising avenues in this regard is the use of electroencephalogram (EEG)-based brain–computer interface (BCI) methods. These technologies provide a way to tap into the brain’s electrical activity that is non-invasive, cost-effective, and versatile [1,2,3,4]. By detecting the neural signals associated with the intention or attempt to move, EEG-based BCIs can be used to facilitate the rehabilitation process. They provide real-time feedback to the user and potentially retrain the brain to regain control over lost motor functions [5,6]. These achievements highlight the potential of EEG-based approaches in restoring motor function and enhancing the quality of life for individuals with impaired upper limbs [7,8,9,10,11]. 

An in-depth study of the relationship between the activation patterns of brain regions during exercise and the neuromuscular control of human movement, i.e., the brain–muscle link, is an important prerequisite for being able to use BCI control [12]. Rahman et al. assessed the correlation between arm exertion and EEG power spectral density in four frequency bands under isometric contraction force, and the results indicated that the associations between the beta EEG frequency bands and different contraction forces of the arm were statistically significant [13]. Rao et al. investigated the relationship between neural variability in frontal, central, and parietal regions and grip strength magnitude in an isometric control task, and their results showed that the sample entropy of the EEG varied regularly at the electrodes in the central region, while there were no significant changes in the frontal and parietal regions [14]. While extensive research has been conducted on the utilization of EEG-based brain–computer interfaces (BCI) for the manipulation of upper limb movements, there is a discernible gap in the literature regarding the precision with which EEG signals can discriminate among the subtle variations in force generated during isometric muscle contractions. Exploring EEG-based methods to assess upper extremity strength can help improve the precision of upper extremity movements. It also has the potential to provide valuable insights into the neural mechanisms related to strength production. This exploration can contribute to the development of targeted strength enhancement and exercise rehabilitation interventions. Slobounov et al. investigated the behavioral and cortical electrical responses of fingers during exertions at 25%, 50%, and 75% of the maximum voluntary contraction (MVC), revealing no statistically significant differences in EEG components [15]. Joseph et al. performed isometric and isotonic lower limb joint movements with eight neurologically intact subjects and utilized adaptive mixed independent component analysis. The classifier had a four-class classification accuracy of 69% compared to 87% for a classifier based on cortical power from multiple brain regions [16]. While initial research has begun to elucidate the interrelationships between EEG and force, there remains a scarcity of studies focusing on the assessment of identical or analogous movements under isometric contraction conditions.

Entropy, in the context of signal processing and EEG analysis, is a measure of the unpredictability or randomness of a signal. It provides a quantitative way to capture the complexity and information content of EEG signals. The study on the performance metrics of the sensory motor rhythm-based brain–computer interface (SMR-BCI) found that spectral entropy has a high correlation with SMR-BCI, achieving an average classification rate of up to 89% in a 40-person experiment [17]. Chen et al. used a classification scheme, and four classes were classified with entropy in the BCI competition data, resulting in a motion classification accuracy of 88.36% [18]. Applications of entropy metrics in the analysis of electroencephalogram (EEG) signals have shown promising results in the diagnosis and treatment of central nervous system disorders. Kannathal et al. applied a variety of entropy to the comparison of EEG data between normal and epileptic subjects with an accuracy of 90% [19]. 

The study’s classification scheme uses fusion entropy as a key feature to categorize the behavioral and electrocortical responses observed during hand-grasping movements. These movements are performed at four specific levels of isometric force: 15%, 30%, 45%, and 60%. The focus of the research is on applying fusion entropy as the primary criterion for distinguishing between these different levels of isometric forces through the analysis of noninvasive EEG data. By incorporating fusion entropy, which includes seven entropies commonly used in electrophysiology, the technique described in this study offers a complete framework for understanding how the brain coordinates the application of different levels of force during hand grasping. This method not only improves our understanding of motor control mechanisms but also creates new opportunities to investigate the neural basis of movement and force adjustment.

## 2. Methods

### 2.1. Subjects

This study involved ten healthy participants (eight males and two females), who were between 21 and 31 years old. None of them had any previous or current hand or wrist disorders. This study was approved by the Institution Research Ethics Board at Tianjin University. All procedures conformed to the Declaration of Helsinki, and all participants signed the informed consent prior to any experimental procedures. Based on the G∗POWER (version 3.1.9.7), the sample size was determined based on the requirements for achieving a medium effect size, with a statistical power of 0.8, an effect size of 0.25, and a significance level set at 0.05.

### 2.2. Experimental Procedure

EEG data were recorded using an array of 64 electrodes, positioned according to the internationally recognized 10–20 system configuration. This electrode placement ensures a thorough mapping of brain activity by covering important areas on the scalp. The EEG signals were captured with the electrodes’ references carefully positioned at the mastoids on both the left and right sides, improving signal clarity and minimizing external noise interference. Additionally, a designated ground electrode was placed at the AFz position on the forehead, serving as a stable reference point to further improve the signal–noise ratio. Electrode impedances were maintained below 10 kΩ, and signal amplification was conducted using a portable wireless EEG amplifier (NeuSen.W64, Neuracle, Changzhou, China) with a sampling rate of 1000 Hz [20,21,22,23].

The experimental setup for participants was designed to optimize comfort and ensure consistent measurement conditions. Participants were seated in a chair that could be adjusted by height, ensuring that each participant could find a comfortable and supportive seating position. This careful focus on ergonomics ensured that the participants’ forearms were naturally positioned on the experimental table, with palms facing upward, in the best posture for the task at hand. They were instructed to hold a grip strength measuring device, which is designed to assess the maximum voluntary contraction (MVC) of grip strength. This assessment involved the participants performing three repetitions of the grip strength exercise, with the aim of establishing their maximum grip strength. The average of these three repetitions was calculated, providing a reliable measure of each participant’s peak grip strength.

Prior to the initiation of the experimental protocol, a preparatory phase was implemented in which 1 min of resting-state EEG data was recorded for each participant. This step is used for establishing a baseline of the resting state. The experimental protocol is structured into distinct stages within each cycle of the activity. Initially, a force-generating phase of 1 s is required to activate the gripping device, ensuring that the device is engaged by applying a continuous force to achieve a specified level of stretch. This phase transitions into a stretch-holding phase, lasting 6 s, during which the participants must sustain the predetermined force level without deviation (Figure 1). Following this exertion, the protocol includes a 1-s ramp down, or relief phase, allowing participants to gradually reduce the force applied. This relief phase smoothly leads into a rest period, characterized by a 10-s duration of inactivity, providing participants with a necessary recovery interval before the commencement of the next cycle. To ensure that participants stayed in optimal condition throughout the experiment, the rest phase was designed to reduce any potential muscle fatigue caused by the stretching exercise. This cycle of stretching and resting was defined as a single trial. The procedure described constitutes a single trial, and participants were required to replicate this protocol ten times, collectively referred to as a set. A rest period of no less than three minutes was set between consecutive sets, ensuring an appropriate recovery time. The t = 0 position in Figure 1 is the starting point of the active segment extraction position.

The experiment was designed to test the participants’ ability to exert force at four distinct levels, specifically 15%, 30%, 45%, and 60% of their assessed maximum voluntary contraction (MVC) [24]. This graduated approach allowed for a nuanced analysis of grip strength and endurance across a spectrum of exertion levels. To facilitate optimal performance and recovery, appropriate intervals were incorporated between each set. These intervals were strategically planned to afford participants ample time to recuperate, thus preventing the accumulation of muscle fatigue that could potentially confound the experiment’s results.

### 2.3. EEG Data Analysis

The EEG data were preprocessed using the EEGLAB toolkit in MATLAB (MathWorks, Inc.,R2016a, Natick, MA, USA) [25], following a structured sequence to improve data quality and computational efficiency. Initially, the sampling rate of the EEG signal was reduced from 1000 Hz to 250 Hz to decrease the data size and improve processing speed. This resampling step happens before filtering to minimize aliasing effects and maintain the integrity of the EEG signal within the required frequency range. Subsequently, the data underwent re-referencing to stabilize the baseline by averaging the signals from bilateral mastoid electrodes to reduce external noise and improve the signal–noise ratio. Following the re-referencing, a Butterworth bandpass filter with a frequency range of 3 Hz to 40 Hz was applied. Finally, the EEG data were segmented, with each segment extending from the onset of the command to 6 s afterward, resulting in a duration of 6 s per segment. 

To ensure data integrity, independent component analysis (ICA) was employed to eliminate noise stemming from sources, such as oculomotor, electromyographic, and cephalometer activities. Elliptic interpolation was employed to rectify certain corrupted channels.

To perform upper limb strength assessment through EEG, this study used the method of multiple entropy fusion, which contains seven entropy metrics that are effective in electrophysiology and rehabilitation, and envelope entropy, which is commonly used for vibrations, as shown in Table 1. The formulas for the eight entropies are in Appendix A.

In this study, fusion entropy features were systematically constructed by calculating eight distinct entropy measures for each of the EEG channels, ranging from F1 to F8. These measures, collectively termed “fusion entropy” (F = [F1, F2, …, F8]), represent a comprehensive feature set designed to encapsulate the diverse informational content of the EEG data across varying levels of isometric force exertion by participants. The use of combined entropy feature sequences with a support vector machine (SVM) classifier aims to analyze the relationship between EEG signal complexity and various levels of exertion. 

To evaluate the effectiveness of the classification framework, the recognition rate is employed as a key metric. The recognition rate measures the accuracy of the SVM classifier in identifying the correct magnitude of the force from the EEG data, providing an objective measure of the classification results’ validity.

### 2.4. Statistical Analysis

Statistical analysis was conducted using the commercial software SPSS, Version 16.0. (SPSS, Inc., 2007, Chicago, IL, USA). The Kolmogorov–Smirnov test was used to check for any deviations from normality in the data. The reporting of outcomes followed the format of mean ± standard deviation (Std).

The Spearman coefficient was calculated to evaluate the correlation between the eight entropy metrics. A two-way [entropy × subject] analysis of variance (ANOVA) was used to study the impact of individual differences among study subjects and the selection of different entropy features on classification accuracy. Differences were considered significant when *p* < 0.05. For post hoc analysis, the Bonferroni correction method was chosen to adjust for multiple comparisons, reducing the risk of type I errors. The study used the Wilcoxon signed-rank test to assess how different experimental methods (two-class versus four-class classification approaches) affected classification accuracy. 

## 3. Results

The data were partitioned into ten subsets based on individual subjects, allocating 70% of the data for training purposes while reserving the remaining 30% for testing within each dataset.

### 3.1. Correlation Analysis among Multiple Entropy Values

The results of the Spearman correlation analysis revealed a notably high correlation of 0.98 between AE and SE in Figure 2. This significant correlation suggests a substantial overlap in the information captured by AE and SE, indicating that their combined inclusion in the fusion entropy feature vector might lead to redundancy.

Based on this finding, approximate entropy (AE) was subsequently omitted from the fusion entropy feature vector. This decision was informed by the need to optimize the feature set for classification purposes, ensuring that each component contributes unique and valuable information to the overall model. The exclusion of AE aims to enhance the discriminative power of the fusion entropy features by minimizing redundancy and focusing on the most informative entropy metrics. 

Fusion entropy was used to perform EEG categorization at rest versus at 15%, 30%, 45%, and 60% MVC force application. The classification accuracy is as follows: 99.94% ± 0.01, 99.93% ± 0.01, 99.99% ± 0.01, and 99.97% ± 0.01 (mean ± standard deviation).

### 3.2. Resting and Active States

The classification performance of the fusion entropy for EEG data was evaluated across different conditions: at rest and during the application of 15%, 30%, 45%, and 60% of MVC force. The accuracy of categorizing EEG states into static (rest) and non-static (force application) was quantitatively assessed. The results are reported as the mean ± standard deviation for each condition, with the following classification accuracies observed: 99.94% ± 0.01 for rest versus 15% MVC, 99.93% ± 0.01 for rest versus 30% MVC, 99.99% ± 0.01 for rest versus 45% MVC, and 99.97% ± 0.01 for rest versus 60% MVC.

These parameters suggested that a total of 45 samples would be necessary to detect statistically significant effects within the scope of our research. In the study, 10 participants were engaged, each conducting 10 distinct action experiments, resulting in a total sample size of 100. This method effectively exceeded the calculated requirement, enhancing the robustness of the statistical analyses.

These findings underscore the efficacy of the fusion entropy approach in discriminating between EEG signals associated with static and varying levels of non-static force application states. The high classification accuracies, consistently exceeding 99.9% across all conditions, indicate that fusion entropy features possess a significant discriminative capability. This suggests that the fusion entropy method is highly effective for categorizing EEG signals into rest and active force application states.

### 3.3. MVC Force Classification Results at 15% and 60% 

In evaluating isometric force exertion in upper extremity muscles, a 15% MVC typically corresponds to the activation of smaller motor units, while more than 60% MVC engages most motor units. The capability of fusion entropy to differentiate between these two levels of force—15% MVC and 60% MVC—is demonstrated with a classification accuracy of 91.73 ± 8.17%, as shown in Figure 3. This performance underscores the effectiveness of fusion entropy features in distinguishing between low and high levels of isometric force.

Classification accuracy using individual entropy metrics yielded the following results: PSE recorded an accuracy of 64.05 ± 10.28%, SSE achieved 67.58 ± 9.84%, LEE marked 71.14 ± 13.14%, SE reached 63.46 ± 9.96%, FE reached at 79.99 ± 11.72%, PE obtained 61.77 ± 10.03%, and EE scored 68.24 ± 16.19%. These outcomes reveal a significant variance in classification accuracy based on the entropy feature selected (F = 10.53, *p* < 0.005, $\eta_{p}^{2}$ = 0.54), highlighting the importance of entropy feature choices for optimal classification effectiveness. Similarly, there was a statistically significant difference in categorization accuracy between subjects (F = 3.65, *p* = 0.01 < 0.05, $\eta_{p}^{2}$ = 0.34), suggesting that changes in accuracy are similarly influenced by individual differences between subjects.

Figure 4 illustrates that, for nine out of ten subjects, fusion entropy yielded high classification accuracies, demonstrating the method’s effectiveness in distinguishing EEG signal patterns. The classification results for these subjects were as follows: 99.38 ± 0.40%; 97.43 ± 0.79%; 93.62 ± 0.78%; 98.33 ± 0.76; 99.80 ± 0.22%; 83.3 ± 1.36%; 78.25 ± 1.49%; 88.95 ± 1.38%; 96.94 ± 0.64%.

Figure 5 shows the confusion matrix under the two classifications, where the sensitivity is 91% and specificity is 91%.

This analysis emphasizes the significance of selecting appropriate entropy features for accurately classifying EEG signals related to different levels of muscle force exertion. The superior accuracy of the fusion entropy method, compared to individual entropy metrics, suggests its robustness as a tool for EEG analysis in the context of motor unit activation and muscle force application.

### 3.4. MVC at 15%, 30%, 45%, and 60%: 4-Level Isometric Contraction Classification Results

Figure 6 presents the classification accuracy obtained using fusion entropy for distinguishing between four different levels of isometric forces (15% MVC and 60% MVC), achieving an accuracy of 69.59% ± 9.66%. This result indicates the method’s capability in a more complex classification task involving multiple force levels, despite the inherent challenges in differentiating between such closely related categories. When individual entropy metrics were employed as eigenvalues for the classification, the outcomes varied significantly as follows: PSE: 36.96 ± 5.29%; SSE: 38.58 ± 5.63%; LEE: 36.66 ± 7.92%; SE: 36.80 ± 5.94%; FE: 49.07 ± 10.30%; PE: 35.13 ± 4.27%; EE: 36.86 ± 8.95%. 

These results further highlight the disparity in classification accuracies when utilizing different entropy features individually, with FE showing a relatively higher performance compared to others. A significant variance in classification accuracy was observed depending on the chosen entropy feature (F = 35.95, *p* < 0.005, $\eta_{p}^{2}$ = 0.80), emphasizing the impact of feature selection on the classification outcome. Similarly to the findings for binary classification, the statistical difference in classification accuracy was noted across subjects (F = 4.75, *p* < 0.005, $\eta_{p}^{2}$ = 0.40), suggesting that the observed variations in performance are attributable to both entropic characteristics and the individual differences between subjects.

Figure 7 shows that for all subjects, fusion entropy achieved the highest classification accuracy compared to individual entropy features: 66.12 ± 1.48%; 77.76 ± 1.50%; 64.31 ± 1.98%; 85.81 ± 0.86; 74.75 ± 1.18%; 70.16 ± 1.41%; 60.01 ± 1.39%; 68.93 ± 1.40%; 76.05 ± 1.08%; 51.99 ± 0.95%.

Figure 8 shows the confusion matrix under the four classifications, as well as the sensitivity and specificity when corresponding to 15%, 30%, 45%, and 60% MVC. The sensitivity of 15% MVC vs. non-15% MVC was 67%, and specificity was 89%; the sensitivity of 30% MVC vs. non-30% MVC was 73%, and specificity was 90%; the sensitivity of 45% MVC vs. non-45% MVC was 65%, and specificity was 90%; the sensitivity of 60% MVC vs. non-60% MVC was 83%, and specificity was 95%.

This analysis underscores the complexity of classifying EEG signals into multiple force levels and the critical role of selecting appropriate entropy features to optimize classification accuracy. Despite the lower overall accuracy in this more challenging task, the fusion entropy method still outperforms individual entropy metrics, illustrating its potential in handling nuanced classification scenarios.

## 4. Discussion

The main conclusions of this study include the following: (1) The eight entropies hold distinct physical interpretations, amalgamating into fusion entropy through mutual supplementation. This fusion demonstrates the capacity to significantly enhance classification efficiency. (2) The adoption of fusion entropy as a feature proves valuable in discerning various isometric contraction forces within the context of the same action, particularly when force divisions are clearly defined (15% MVC and 60% MVC). (3) When fusion entropy is used to classify force, there is no statistical difference in the classification accuracy of different subjects.

Different types of entropy measures offer various perspectives on the signal’s structure and irregularity. Take, for instance, PSE [28,29], which operates as an information entropy designed to quantify the complexity of spectral patterns within the frequency domain from an energy-based perspective. PSE delves into the spectral makeup of EEG signals. A more intricate EEG signal translates into a more uniformly distributed energy across the frequency spectrum, resulting in higher PSE values. On another note, fuzzy entropy finds application in characterizing and analyzing diverse biomedical signals, including EMG, EEG, gait, and heart rate variability [35]. It should be emphasized that simply increasing the number of entropy indicators does not necessarily improve classification accuracy. Simply incorporating many entropy features without a strategic approach or understanding of their relevance can lead to suboptimal outcomes. Certain entropy types are calculated in very similar ways and have the same physical meaning. Examples are approximate entropy and sample entropy. Approximate entropy exhibits an inherent regularity bias due to its self-matching nature, lacks relative consistency between approximate entropy values calculated with different combinations of parameters, and is sensitive to the length of the dataset [38,39]. In contrast, sample entropy overcomes the drawbacks of approximate entropy by avoiding the self-matching of vectors, showing good relative consistency, and it is not affected by the length of the dataset [39]. From a biomedical standpoint, this finding resonates with the intricate nature of neural signals. The brain’s complexity is manifested through multifaceted patterns of neural activity. The fusion of entropies capitalizes on these intricacies, effectively harnessing the richness of information embedded within different entropic measures. This approach aligns with the pursuit of more nuanced and accurate characterizations of neural responses, offering potential applications in areas like neurorehabilitation and brain–computer interfacing.

The results of this study also show that fusion entropy can be used as an effective feature to recognize the same action under different isometric forces. Traditionally, the regulation of force during movement has been closely associated with EMG techniques. However, stroke patients often struggle to control the forces in their arms due to physical changes like muscle weakening, which greatly affects their ability to adjust forces efficiently. In these cases, using EMG directly for identifying and classifying action forces is challenging. Recognizing and controlling these forces are crucial elements of the rehabilitation process, directly impacting the recovery of fine motor movements.

Our investigation contributes to the current understanding of EEG signal analyses in motor control, aligning with and expanding upon findings from previous research. Mahjabeen Rahman’s work highlights how physical conditions, specifically isometric contraction forces, influence brain activity, similarly to our observations of EEG signal variations [13]. Nishant Rao’s findings on central brain region activities in response to varying MVC levels also parallel our discovery of differentiated EEG responses to force magnitudes [14]. Furthermore, our results support Hendrik Enders’ discussions on EEG-EMG connectivity, suggesting integrated muscular and cortical activities during motor control [12]. Our study employed entropy metrics to analyze EEG signals for discriminating between force levels in isometric contractions. This allows for a broader quantification of cortical muscle connectivity that is sensitive to variations in force, a nuance not specifically addressed in Rao’s findings regarding activity in frontal and parietal lobes. Moreover, while existing studies emphasize the significance of certain brain regions or the effects of comfort levels on motor execution, our research demonstrates that entropy metrics provide a novel means to quantify force exertion magnitudes, enhancing the analysis of EEG signal variability associated with motor tasks. Our analysis elucidates the control mechanisms underlying motor execution. Through entropy measures, we offer evidence of the brain’s intricate involvement in modulating force levels, enriching the understanding of cortical participation in motor control.

In addressing the complexities of force control and recognition in stroke rehabilitation, EEG emerges as a particularly promising avenue. Unlike traditional methods that predominantly focus on peripheral measurements like EMG [40,41], EEG delves into the central nervous system’s perspective. It allows for the capture of motor intentions directly from the brain’s electrical activity. For stroke patients, this holds immense potential. It enables them to bypass the hurdles posed by compromised peripheral pathways and communicate their motor intentions directly through neural signals. Furthermore, EEG does not just stop at recognizing motor intentions. It also plays a pivotal role in motor execution. By tapping into the brain’s electrical patterns, EEG-based methods facilitate the translation of these intentions into motor commands. This aspect is particularly valuable for stroke rehabilitation, where the re-establishment of effective motor execution is a central objective. The ability of electroencephalography to recognize motor intent and enable motor execution offers a promising paradigm shift in addressing the complex challenges faced by stroke patients in force control and fine motor movements. 

This study presents certain limitations that warrant consideration for future research. A notable limitation of our current approach is the use of SVM classifiers, which, while effective for our purposes, limited our ability to distinguish the contributions of different EEG channels in the classification process. Consequently, the effects of specific brain regions on motor actions were not examined in detail. This limitation highlights the need for future studies to employ analytical methods that can more accurately attribute classification weight to individual channels, thereby offering a deeper understanding of the regional brain dynamics involved in motor control. Furthermore, the study’s categorization into four force levels, although sufficient for our initial objectives, may not fully capture the nuances of fine motor movements. A more granular categorization could potentially reveal additional insights into the complex nature of motor control, especially in tasks requiring subtle force adjustments.

Another limitation is the focus on healthy subjects, which provided a controlled environment for initial investigations. However, the inclusion of populations with motor impairments, such as stroke patients, in future studies could significantly enhance the applicability and impact of our findings. Understanding how entropy-based features of EEG signals vary among individuals with motor deficits could inform rehabilitation strategies and contribute to the development of brain–computer interfaces tailored to assistive technologies. In response to these limitations, our future work will aim to incorporate a broader array of entropy metrics, employ advanced classification techniques to better understand the role of different brain regions, and expand our categorization of force levels. Additionally, by extending our research to include individuals with motor impairments, we hope to contribute to the broader application of our findings in clinical and rehabilitative settings. 

Finally, the fusion of entropy measures as a feature did not statistically differ in the classification of different isometric contractility across subjects. This observation assumes particular significance given that electroencephalography (EEG) measurements are notably affected by individual variations. This consistency in fusion entropy’s classification performance, irrespective of individual differences, underscores its potential utility as a dependable feature in broader applications. Given the inherent variability in EEG signals due to subject-specific factors, the consistent performance of fusion entropy lends support to its validity for widespread implementation in tasks involving force classification. This finding has implications for enhancing the reliability and generalizability of EEG-based classification methodologies across diverse individuals.

## 5. Conclusions

The meticulous control of force magnitude emerges as a pivotal element in the synergy of refined upper limb motions. Different from traditional EMG methods used to classify force, this paper uses fusion entropy as the feature to classify EEG into two categories, specifically 15% and 60% maximum voluntary contraction (MVC). It also utilizes fusion entropy to classify voluntary contraction into four specific categories: 15%, 30%, 45%, and 60% MVC. The results show that the fusion entropy feature achieves high classification accuracies in all cases. Additionally, fusion entropy was not statistically differentiable as a feature in categorizing the separate isometric forces of the different subjects. The reliable performance of fusion entropy, despite the wide variation in EEG signals due to individual characteristics, adds to the evidence that it is suitable for broad use in tasks related to force detection. Using electroencephalography (EEG) to determine motor intent and facilitate motor performance offers a promising approach to addressing the complex challenges faced by stroke patients in controlling force and fine motor movements.

## Figures and Tables

**Figure 1 sensors-24-02323-f001:**
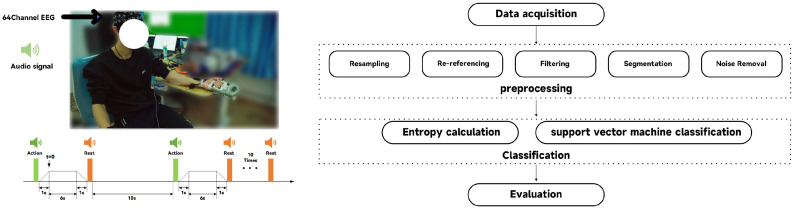
EEG experimental processes and data processing.

**Figure 2 sensors-24-02323-f002:**
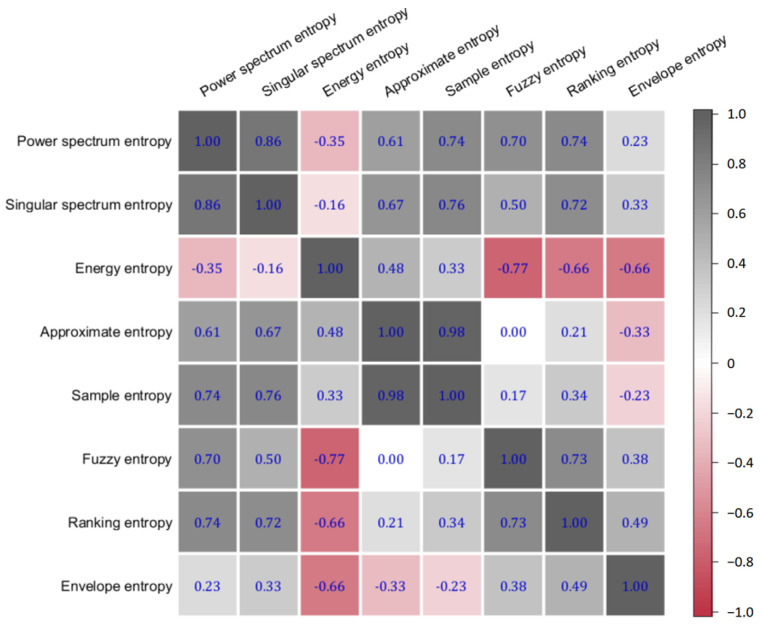
Spearman coefficient between different entropies.

**Figure 3 sensors-24-02323-f003:**
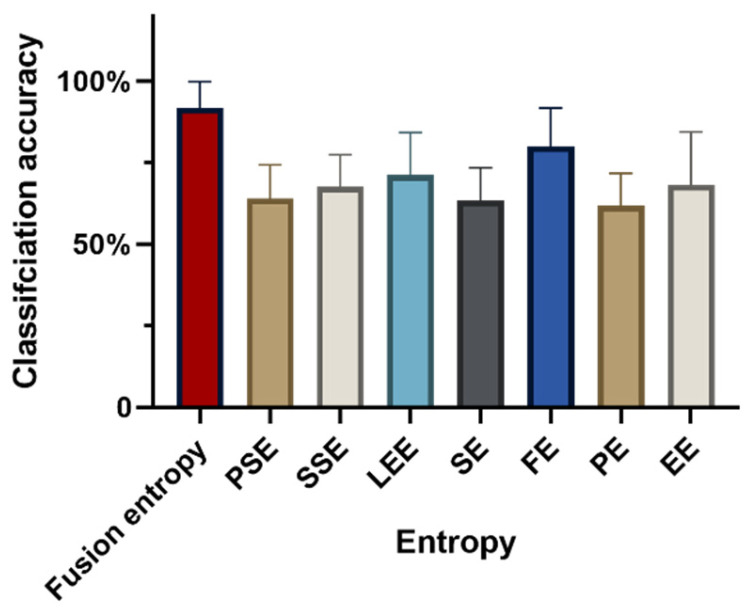
Accuracy of classification using different entropies as features for 15% and 60% MVCs.

**Figure 4 sensors-24-02323-f004:**
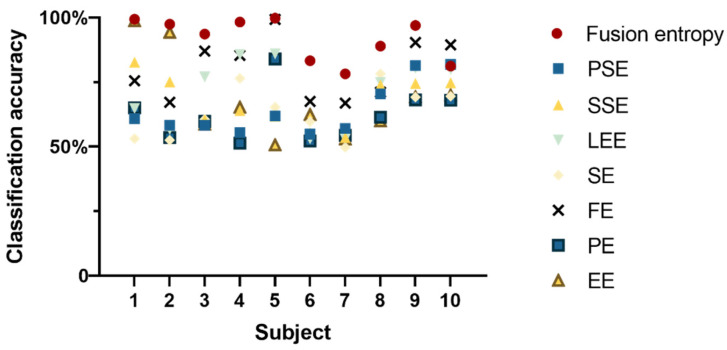
Accuracy of entropy feature classification for different subjects at 15% and 60%.

**Figure 5 sensors-24-02323-f005:**
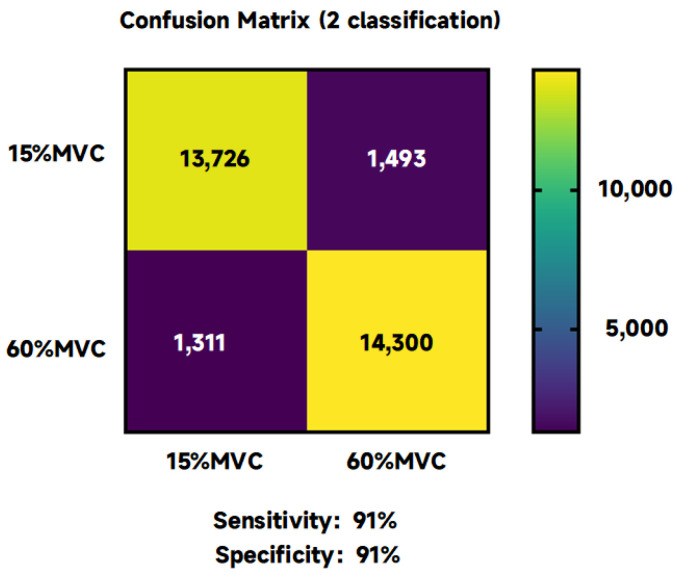
Two classification confusion matrixes: sensitivity and specificity.

**Figure 6 sensors-24-02323-f006:**
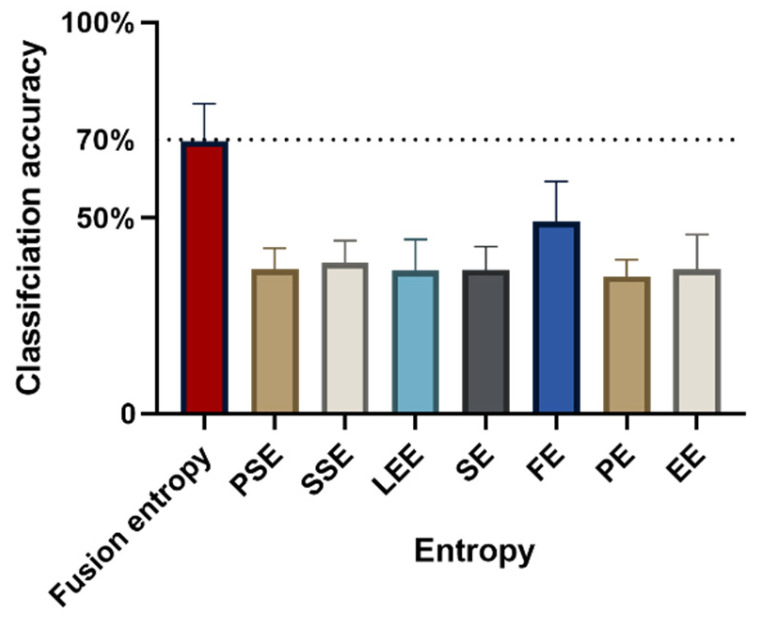
Accuracy of classifying 4-level isometric contractions using different entropies as features. The dotted line indicates the 70% level of classification accuracy.

**Figure 7 sensors-24-02323-f007:**
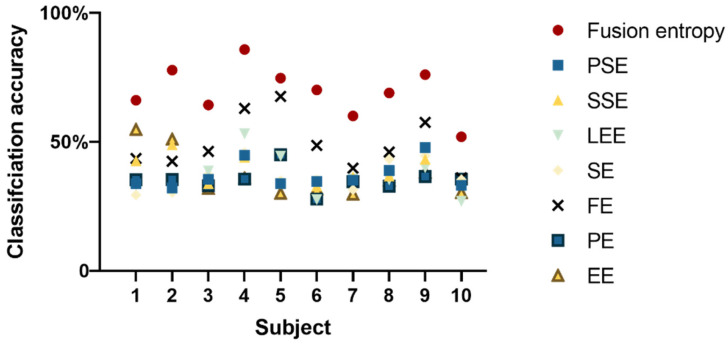
Four-level isometric force classification results.

**Figure 8 sensors-24-02323-f008:**
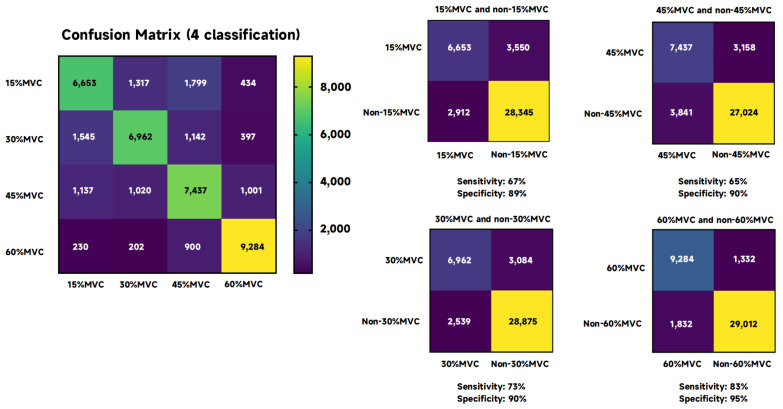
Four confusion matrix classifications and the sensitivity and specificity of each contractile force.

**Table 1 sensors-24-02323-t001:** Entropy metrics and interpretability in multiple entropy fusion.

Entropy Name	Interpretability	Bibliography
power spectral entropy (PSE)	PSE quantifies signal complexity by measuring power distribution uniformity across frequencies; higher values indicate greater complexity and disorder.	[26,27,28,29]
singular spectral entropy (SSE)	SSE measures the complexity of the data by analyzing the variance distribution of the eigenvectors from the singular value decomposition; higher entropy values indicate a more homogeneous and complex distribution.	[30]
log energy entropy (LEE)	LEE can characterize the complexity of EEG sub-bands and help successfully classify EEG data by quantifying the variability of signal energy distribution in different frequency bands.	[31,32]
approximation entropy (ApEn)	ApEn quantifies EEG time series regularity, handling stochastic components. Low ApEn indicates predictability; high ApEn suggests uncertainty.	[33,34]
sample entropy (SE)	SE, as a modified form of ApEn, is used to assess the complexity of any physiological signals, including the EEG. More generally, it shows stability, reducing the bias of ApEn, and it is independent of the signal length.	[35]
fuzzy entropy (FE)	FE measures similarity in biomedical signals like electromyography (EMG), EEG, and heart rate variability, using a fuzzy function to analyze time series signals with N sample lengths.	[26]
permutation entropy (PE)	PE assesses time series complexity by comparing neighboring values, mapping to ordinal patterns, identifying non-linear signals, reducing problem space, and enhancing noise robustness.	[28,36]
envelope entropy (EE)	Envelope entropy, used in signal processing, measures entropy by quantifying the regularity or complexity of time-series data, offering insights into signal characteristics.	[37]

## Data Availability

Data are contained within the article.

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
