# Peer review of "The EEG-Based Fusion Entropy-Featured Identification of Isometric Contraction Forces under the Same Action"

_sensors, 2024, doi:10.3390/s24072323_

Round 1
Reviewer 1 Report
Comments and Suggestions for Authors
The manuscript addresses the topic of recent interest motivated by the development of brain-computer interfaces and diagnostic tools. The principal attractive feature of such schemes is their non-invasiveness. At the same time, they require more complicated data processing and search for reliable indicators. Various definitions of entropy are used for this goal. The authors accurately and statistically correctly evaluate these measures. Thus, the obtained results are of interest and the manuscript may be accepted in principle but only after substantial revision aimed at improving its readability and completeness of result presentation.
Major issues suggested for improvement:
1 1. The complete characterization of classification schemes should include not only accuracy but also sensitivity, specificity, etc. forming a full confusion matrix. It should be calculated and discussed.
- A multichannel scheme is used in this work but the description of the channel selection and usage is not very clear in the present state of the text: the authors mention 8 channels while there is a 64-electrode scheme; further simple “EEG data” are mentioned without particular explanations. The procedure of the data selection and processing of multichannel data should be described in more detail.
- Additionally, it is known that motor activity differently affects signals from differently placed electrodes, including the entropic features, see e.g. Pitsik, E., Frolov, N., Hauke Kraemer, K., Grubov, V., Maksimenko, V., Kurths, J., & Hramov, A. Motor execution reduces EEG signals complexity: Recurrence quantification analysis study. Chaos: An Interdisciplinary Journal of Nonlinear Science, 30 (2020) 023111. This specificity should be checked and analysed respectively to the present experiment.
Minor points:
4. The author often uses too pretentious words, for example:
Line 113: “using a sophisticated array of 64 electrodes” What is especially sophisticated in this scheme? It is a conventional placing;
Line 130: “This device, renowned for its precision and reliability” looks like advertising
Line 133 : “capture of a broad spectrum of EEG frequencies with exceptional detail and accuracy”. Again it is pretentious wording; it one needs to highlight the specificity of the measurement accuracy, quantitative measures should be provided instead of buzzwords
5. Although the meaning of used entropies is described in the table, it is advisable to provide mathematical definitions in the Appendix and comment on, which data go as inputs.
Comments on the Quality of English Language
See the report (the subsection "Minor issues")
Author Response
Thank you for your detailed feedback. I have addressed each of your comments positively, with specific responses and revisions detailed in the attached document. I appreciate your guidance and hope my revisions meet your expectations.

Reviewer 2 Report
Comments and Suggestions for Authors
“EEG-Based Fusion Entropy-Featured Identification of Isometric Contraction Forces Under the Same Action”(sensors-2842018)
This manuscript aimed to use multiple entropy features to differential different level of isometric contraction forces under the same action. The results revealed that including a wide range of entropy features significantly improves classification efficiency; and the fusion entropy method shows exceptional accuracy in classifying isometric contraction forces. Overall, this topic is interesting and the findings might hold great practical implications. However, some concerns appeared after reading the whole manuscript.
1. The current literature review seems rather limited and did not cover all the related papers. Some important papers need to be reviewed and discussed (please carefully check all the references mentioned in the following papers):
Rahman, M., Karwowski, W., Sapkota, N., Ismail, L., Alhujailli, A., Sumano, R. F., & Hancock, P. A. (2023). Isometric Arm Forces Exerted by Females at Different Levels of Physical Comfort and Their EEG Signatures. Brain Sciences, 13(7), 1027.
Rao, N., Paek, A. Y., Contreras-Vidal, J. L., & Parikh, P. J. (2023). Lateralized Neural Entropy modulates with Grip Force during Precision Grasp. bioRxiv, 2023-05.
Enders, H., & Nigg, B. M. (2016). Measuring human locomotor control using EMG and EEG: Current knowledge, limitations and future considerations. European journal of sport science, 16(4), 416-426.
After providing the appreciate literature review, please clearly identify the current research gaps and the potential novelties of current investigation and discuss the current findings with the most-related previous ones. The current discussion is too superficial and it seems no related previous studies exist.
2. How did you determine the sample size? Did you calculate the sample size needed before formal study?
The current sample size seems too little to get reliable results.
3. Please provide the validation data or paper about the EEG amplifier you used (portable wireless EEG amplifier (NeuSen.W64, Neuracle, China)).
4. SVM should be explained in the title of figure 1.
5. Line 189-195 replicate part of the content of the same paragraph, please carefully re-check all the context to ensure to be concise.
6. EEGLAB needs reference.
Delorme, A., & Makeig, S. (2004). EEGLAB: an open source toolbox for analysis of single-trial EEG dynamics including independent component analysis. Journal of neuroscience methods, 134(1), 9-21.
7. Abbreviations need to be defined the first time they appear in the text, and please write the full name and give the abbreviation in parentheses. However, some abbreviations with full names appeared at the second time, please carefully re-check that.
8. Figure 3 is not necessary and does not provide additional information.
9. Please provide the effect size where available.
10. Some parts in the current discussion need to be moved to the introduction part to justify the analysis method you choose.
11. I strongly recommend that the paper be thoroughly proofread and edited for languages and grammars, to enhance readership, especially the logic and thus, the introduction and discussion part need to be reorganized.
Comments on the Quality of English Language
Extensive editing of English language required
Author Response

(The authors gave the same response as above.)

Reviewer 3 Report
Comments and Suggestions for Authors
The paper “EEG-Based Fusion Entropy-Featured Identification of Isometric Contraction Forces Under the Same Action” investigates the application of fusion entropy from EEG signals to differentiate isometric contraction forces in upper limb movements, aiming to enhance human-computer interface control. By integrating eight different entropy measures, the study achieved significant advancements in classification efficiency, demonstrating a high accuracy rate of 91.73% in distinguishing between 15% and 60% maximum voluntary contraction forces, and a 69.59% accuracy across four force levels. These results highlight the potential of fusion entropy in EEG signal analysis for precise isometric contraction detection, offering new avenues for motor control and rehabilitation technologies.
Section 1. Introduction
1) Overall, the manuscript is highly relevant in the metrics aspect. Throughout the document, several sentences are repetitive and should be carefully checked. Be as insightful and precise as possible.
2) Line 62, page 2. “with impaired upper [7–11].” Something is missing here. Upper limbs?
3) Line 64, page 2, “widely used to classify patterns based on EEG. Zhang et al.” Reference seems to be broken. Zhang et al. is between two sentences. Probably referring to a citation for the preceding sentence.
4) Introduction lines 64 to 71, instead of enumerating references and findings, they should be addressed in a coherent paragraph, highlighting the contributions and important findings in a way that blends in with the narrative being told in previous sentences.
5) Paragraph of lines 89-92, and table 1 should be addressed in the methods section, probably under 2.3. EEG Data Analysis section. Finishing the introduction section with the paragraph of lines 94-104, where the aims and focus of the study are presented.
6) The sentences “Electrode impedances were maintained below 10 kΩ, and signal amplification was conducted using a portable wireless EEG amplifier (NeuSen.W64, Neuracle, China) with a sampling rate of 1,000 Hz.” of lines 123-125 are repeated in the sentence “To ensure the integrity and reliability of the EEG recordings, we rigorously checked and maintained the electrode impedance below 10 kΩ. Signal amplification was expertly done using a cutting-edge portable wireless EEG amplifier, the NeuSen.W64 model from Neuracle in China. This device, renowned for its precision and reliability, was configured with a high sampling rate of 1,000 Hz.” From lines 127-131.
7) Throughout the entire document and almost all sections, repeating sentences like previous comments are present.
Section 2.2. Experimental Procedure
8) The experimental methodology is not clear. Paragraph from line 147: “At the beginning of the experiment, participants were told to start the grip device by applying a continuous force to reach a set level of stretch. This activation needed to be maintained for exactly one second, initiating a stretching phase that lasted for a total of six seconds. After completing the exertion phase, participants were required to enter a rest phase, lasting at least 10 second”. Is it one second or six seconds? What about the extra second of the ramp-down of the figure 1.
Section 2.3. EEG Data Analysis:
9) lines 190 to 193, “To enhance computational efficiency, the EEG signal was resampled from the original 1000 Hz to 250 Hz. Re-referencing was achieved using bilateral mastoid averages. A Butterworth bandpass filter, with a range of 3 Hz to 40 Hz, was applied.” Is repeated from sentences in lines 178-179 and 184-185.
10) Data segmentation is not clear. “Finally, the EEG data were segmented, with each segment extending from the onset of the command to six seconds afterward, resulting in a duration of five seconds per segment.” According to Figure 1, from onset (start contraction sound), there is depicted a one-second contraction ramp-up, then a stable contraction hold for six seconds, and a one-second ramp down to the rest stage. From onset to six seconds afterwards, there should be six seconds. A diagram of segmentation procedure could clarify this. Please reference “Action Onset as t=0”.
11) One of the first presented results is that the classifier is able almost perfectly to discriminate contraction segments with respect to rest segments. How are the “rest” segments selected, and from which timeframe of the experiment are they from between trials? Are they also 5 seconds as the force exertion segments? Please add this to the methods.
12) How many static and non-static segments were classified for the given accuracy results of 99.94%±0.01 for rest versus 15% MVC, 99.93%±0.01 for rest versus 30% MVC, 99.99%±0.01 for rest versus 45% MVC, and 99.97%±0.01 for rest versus 60% MVC.
13) 64-channel EEG is used to compute 8 entropy-based features for each channel. Please provide more information on what EEG channels are involved in the grip strength information. Are electrodes on the Primary Motor Cortex (M1), supramarginal gyrus (SMG), ventral premotor cortex (PMv), and somatosensory cortex (S1) involved in the discriminative power of entropy-based features?
14) A topological map Figure showing features relevance would highly enhance the contribution, linking the information provided by the entropy fusion metrics and the topological characterization for Brain-machine interfaces.
15) The most important missing information in the provided manuscript is regarding the fusion entropy itself. Table 1 describes the different entropy measurements and metrics implemented in the EEG signals. However, no information is given for the fusion algorithm. How are the features combined?
2.14.0.0 2.14.0.0 Comments on the Quality of English LanguageEnglish should be checked as several of the paragraphs repeat information over and over again in different sections. The manuscript should improve the structure and check the information given in each paragraph.
2.14.0.0 2.14.0.0Author Response
Thank you for your detailed feedback. I have addressed each of your comments positively, with specific responses and revisions detailed in the attached document. I appreciate your guidance and hope my revisions meet your expectations.

Round 2
Reviewer 1 Report
Comments and Suggestions for Authors
The authors significatntly impoved the manuscript making is more clear and readable; the main message and results are well-denoted now. I recomend acceptance of this work.
Author Response
Thank you very much for your encouraging feedback and for recognizing the improvements made to the manuscript. Your comments were invaluable in enhancing its clarity and readability. We are grateful for your recommendation for acceptance.
Reviewer 2 Report
Comments and Suggestions for Authors
1. The determination of sample size should be moved to “2.1. Subjects” part. What tool did you use to determine the sample size, and please provide the screenshot of the determination process in the response letter.
2. About the “2.4. Statistical Analysis”,please provide the specific factor and the related level for the ANOVA analysis.
3. You may misunderstand my concern about effect size. Please provide the effect size for your ANOVA analysis, such as η².
Comments on the Quality of English LanguageModerate editing of English language required
Author Response
Thank you very much for guiding us with your insightful feedback on the statistical aspects of our manuscript. We have diligently implemented the suggested changes and have also included screenshots to clearly address and resolve any concerns you may have had. Detailed responses and the corresponding documentation are provided in the attachment for your review. We are grateful for your assistance in enhancing the quality and integrity of our work.

Reviewer 3 Report
Comments and Suggestions for Authors
Dear authors, all my previous comments were addressed in text modifications, enhancing the overall text quality. Also, aspects not directly integrated were addressed as limitations and future work. The introduction, methods, discussion, and results have been clearly enhanced.
The application and results are encouraging for entropy-based metrics in the field. I encourage the authors for their contribution and to continue for future work results with higher spatial resolution, more emphasis on brain regions and proof of concept on patients with motor impairments.
2.14.0.0Author Response
Thank you for recognizing the enhancements made to the manuscript and for your encouraging remarks regarding our work on entropy-based metrics. Your guidance has been instrumental in refining our text and in framing our limitations and future research directions more clearly. We are inspired by your suggestions to pursue further studies with higher spatial resolution, a focused analysis on specific brain regions, and application to patients with motor impairments. We look forward to contributing more to this field, building upon the foundation this study has established.